# PeerJ

# Influence of coral and algal exudates on microbially mediated reef metabolism

Andreas F. Haas[1,2], Craig E. Nelson[3], Forest Rohwer[1],
Linda Wegley-Kelly[1], Steven D. Quistad[1], Craig A. Carlson[3,4],
James J. Leichter[2], Mark Hatay[1] and Jennifer E. Smith[2]

[1] Department of Biology, San Diego State University, United States
[2] Scripps Institution of Oceanography, University of California, San Diego, United States
[3] Marine Science Institute, University of California, Santa Barbara, United States
[4] Department of Ecology, Evolution and Marine Biology, University of California, Santa Barbara, United States

Corresponding author
Andreas F. Haas,
andreas.florian.haas@gmail.com

## ABSTRACT

Benthic primary producers in tropical reef ecosystems can alter biogeochemical cycling and microbial processes in the surrounding seawater. In order to quantify these influences, we measured rates of photosynthesis, respiration, and dissolved organic carbon (DOC) exudate release by the dominant benthic primary producers (calcifying and non-calcifying macroalgae, turf-algae and corals) on reefs of Moʻorea French Polynesia. Subsequently, we examined planktonic and benthic microbial community response to these dissolved exudates by measuring bacterial growth rates and oxygen and DOC fluxes in dark and daylight incubation experiments. All benthic primary producers exuded significant quantities of DOC (roughly 10% of their daily fixed carbon) into the surrounding water over a diurnal cycle. The microbial community responses were dependent upon the source of the exudates and whether the inoculum of microbes included planktonic or planktonic plus benthic communities. The planktonic and benthic microbial communities in the unamended control treatments exhibited opposing influences on DO concentration where respiration dominated in treatments comprised solely of plankton and autotrophy dominated in treatments with benthic plus plankon microbial communities. Coral exudates (and associated inorganic nutrients) caused a shift towards a net autotrophic microbial metabolism by increasing the net production of oxygen by the benthic and decreasing the net consumption of oxygen by the planktonic microbial community. In contrast, the addition of algal exudates decreased the net primary production by the benthic communities and increased the net consumption of oxygen by the planktonic microbial community thereby resulting in a shift towards net heterotrophic community metabolism. When scaled up to the reef habitat, exudate-induced effects on microbial respiration did not outweigh the high oxygen production rates of benthic algae, such that reef areas dominated with benthic primary producers were always estimated to be net autotrophic. However, estimates of microbial consumption of DOC at the reef scale surpassed the DOC exudation rates suggesting net consumption of DOC at the reef-scale. *In situ* mesocosm experiments using custom-made benthic chambers placed over different types of benthic communities exhibited identical trends to those found in incubation experiments. Here we provide the first comprehensive dataset examining direct primary producer-induced, and indirect microbially mediated

alterations of elemental cycling in both benthic and planktonic reef environments over diurnal cycles. Our results highlight the variability of the influence of different benthic primary producers on microbial metabolism in reef ecosystems and the potential implications for energy transfer to higher trophic levels during shifts from coral to algal dominance on reefs.

## INTRODUCTION

Coral reefs, although generally located in oligotrophic environments, are one of the most biodiverse ecosystems on the planet, due largely to their high productivity and efficient nutrient recycling mechanisms (*Done et al., 1996*). In recent years coral reefs have suffered many impacts, from local anthropogenic influences such as pollution, fishing, and coastal development, to global climate change including warming and likely future acidification of the oceans (*Hoegh-Guldberg et al., 2007*). In many areas, these impacts have led to changes in community structure often resulting in a phase shift (*McCook, 1999*), in which the benthic communities shift from dominance by corals to fleshy algae (*McManus & Polsenberg, 2004*, Smith et al., unpublished data). These shifts in benthic community structure have dramatic implications for the overall trophic structure of tropical reefs as corals provide habitat and shelter for numerous taxa (*Paine, 1980*; *Alongi, 1994*). In addition, different benthic primary producers influence reef communities in multiple, distinct ways by regulating the availability of various inorganic and organic resources and altering the physical structure of the benthos (*Done, 1992*; *Wild et al., 2011*).

Organic material supplied to the ecosystem by benthic primary producers as exudates is thought to play a pivotal role in community-wide transitions on coral reefs (*Jones, Lawton & Shachak, 1997*; *Wild et al., 2004a*). Exudates may serve different ecological functions depending on their origin. Coral exudates may keep valuable resources in oligotrophic reef systems by trapping particles from the water column, which are remineralized by the benthic microbial communities. In contrast, algae derived exudates have been shown to stimulate rapid growth of planktonic microbes (*Haas et al., 2011*) and community shifts towards copiotrophic and potentially pathogenic microbial communities in the water column (*Nelson et al., 2013*). Despite an increasing focus on microbial cycling of carbon in recent years, little is still known about the variability in microbial uptake of dissolved organic matter (DOM) and the potential influence it may have on biogeochemical cycling.

Previous studies of tropical reef-associated primary producers have shown that all primary producers release a significant portion of their photosynthetically fixed carbon immediately into their environment (*Crossland, 1987*; *Ferrier-Pagès et al., 1998*; *Wild et al., 2010*). It has further been established that fleshy macroalgae and especially small (<2 cm) filamentous algal turfs generally have noticeably higher DOC release rates than calcifying primary producers including hermatypic corals. These results are consistent over a wide

range of reef systems, comprising the Red Sea (*Haas et al., 2010a*; *Naumann et al., 2012*), the Caribbean (*Haas et al., 2010b*), and the Central Pacific (*Haas et al., 2011*).

However, counter to expectations, *Nelson et al. (2011)* demonstrated that in a backreef system dominated by algae rather than corals, DOC concentrations were significantly lower than in the surrounding offshore waters. Other studies (*Dinsdale et al., 2008*) incorporating multiple islands in the central Pacific have shown similar patterns where fleshy algal abundance is inversely related to DOC concentrations in the water column. This surprising inverse correlation may be explained by a significantly more heterotrophic microbial metabolism following initially higher availability of algae derived bio-available DOC. A system wide decrease in DOC concentrations could then be the result of (a) increases in the abundance of heterotrophic microbes (*Dinsdale et al., 2008*) and, (b) a co-metabolism, which occurs when microbes are given an initial surplus of labile carbon, enabling this bacterial community to utilize refractory carbon sources (*Carlson et al., 2002*).

Recent research has shown that macroalgae derived exudates, enriched in the dissolved combined neutral sugar components Fucose and Galactose, facilitate significantly higher rates of bacterioplankton growth and concomitant DOC utilization than coral exudates or untreated seawater (*Haas et al., 2011*; *Nelson et al., 2013*). Further, microbial communities growing in different exudates selectively remove different dissolved combined neutral sugar (DCNS) components, whereby the bacterial communities growing on algal exudates have significantly higher utilization rates of the sugar components which were enriched in the respective algal exudates. Analysis of microbial community composition identifies clear differentiation between the communities selected for by algae exudates and those growing on coral exudates or seawater controls. Macroalgae fostered rapid growth of less diverse communities and selected for copiotrophic bacterial populations with more opportunistic pathogens – so called super-heterotrophic communities (*Nelson et al., 2013*; *Dinsdale & Rohwer, 2011*). In contrast coral exudates engendered a smaller shift in bacterioplankton community structure and maintained relatively high diversity.

The microbial landscape on tropical reefs, however, is not only restricted to the water column directly adjacent to the reef benthos ($\sim 10^5$–$10^6$ cm$^{-3}$) (*Azam, 1983*; *McDole et al., 2012*). In addition to microbes associated with benthic macro-organisms ($>10^7$ cm$^{-2}$ surface area) (*Rosenberg et al., 2007*), those associated with calcareous reef sands ($\sim 10^9$ cm$^{-3}$) (*Schöttner et al., 2011*; *Hansen et al., 1987*; *Sørensen et al., 2007*; *Rusch, Hannides & Gaidos, 2009*) and the vast porous reef structures in the reef matrix (*de Goeij et al., 2008*) may also play a significant role in biogeochemical cycling (*Capone et al., 1992*; *Wild et al., 2005*; *Werner et al., 2008*; *Scheffers et al., 2004*). Surface associated microbes may carry out multiple ecological functions, such as nitrogen fixation or inhibition of potential pathogens (*Brown & Bythell, 2005*) for their host organisms. The benthic microbial communities, living in the reef structure or reef sands, on the other hand have been recognized as important components for the reef community, as they are capable of rapidly reallocating nutrients in the otherwise oligotrophic tropical reef environments (*Rasheed et al., 2004*). They also may constitute an essential food source for

protists and invertebrates, forming the base of benthic food webs (*Alongi, 1994*). Next to remineralization and redistribution of nutrients, recent studies have emphasized the role of the benthic microbial communities as important primary producers in these ecosystems (*Boucher et al., 1998*; *Clavier & Garrigue, 1999*; *Heil et al., 2004*; *Werner et al., 2008*).

Like the planktonic microbial communities, benthic microbes, although significantly different in community composition (*Hewson & Fuhrman, 2006*), are also known to be affected by primary producer derived exudates (*Sjöling et al., 2005*; *Wild et al., 2004b*). However, their responses to these different types of organic matter have rarely been investigated (*Wild et al., 2005*) and only one pilot study addressed these questions in the context of changing reef environments (*Wild et al., 2008*). Additionally, production, respiration and the contribution to the nutrient pool have been assessed independently for all of the above described groups, but their relative contribution to collective reef metabolism and their effects on each other have not been investigated on a community scale. Although there have been previous attempts to quantify primary production budgets in coral reefs (e.g., *Odum, 1968*; *Gordon, 1971*; *Sournia, 1976*; *Hatcher, 1990*) they did not account for how microbial metabolism associated with different reef organisms may influence these processes at a landscape scale (*Hoegh-Guldberg et al., 2007*; *Hughes et al., 2007*).

The goal of the present study was to quantify biochemical processes and metabolic rates of both benthic and planktonic microbial communities across different benthic assemblages over diurnal cycles. To verify the findings of our controlled incubation experiments we simultaneously conducted a series of *in situ* measurements, using collapsible benthic isolation tents (cBITs), which allow for continuous monitoring of enclosed portions of the reef benthos. These data present the first comprehensive assessment of how benthic primary producers influence surrounding seawater chemistry directly through metabolic processes and indirectly via changing the microbial landscape and metabolism in this ecosystem. Our study thus provides a system-wide overview of potential biochemical alterations facilitated by different primary producer communities on coral reefs.

## MATERIAL AND METHODS

### Study site

This study was conducted at the Richard B. Gump South Pacific Research Station located on the north shore of the island of Moʻorea, French Polynesia (17.48 S 149.84 W) from 1 to 22 September 2011. The high volcanic island of Moʻorea is encircled by a barrier reef approximately 1 km offshore, thereby creating a semi-enclosed backreef system. The reef ecosystem consists of an outer reef slope and a lagoon system comprising a backreef platform with average water depths of 3 m, and a fringing reef bordering the island (*Hench, Leichter & Monismith, 2008*; *Nelson et al., 2011*). The benthic community on the backreef platform, our main area of investigation, is composed of approximately $68.6 \pm 4.9\%$ turf- and fleshy macroalgae, $22.7 \pm 4.4\%$ hermatypic coral and $8.1 \pm 2.0\%$ sand (http://mcr.lternet.edu/data). Average daytime (06:00 to 18:00 h) PAR availability in the backreef study area during the entire study period was $\sim$580 μmol quanta m$^{-2}$ s$^{-1}$

as measured in LUX in a 5 min resolution with HOBO® Pendant UA-002-64 light and temperature loggers at the water depth were organisms were collected (2.0–2.5 m). Lux were converted to µmol quanta m$^{-2}$ s$^{-1}$ PAR according to the approximation established by *Valiela (1984)*: 1 mmol quanta m$^{-2}$ s$^{-1}$ 400–700 nm = 51.2 LUX). Average backreef *in situ* water temperature at this depth was 26.8 ± 0.6°C with diurnal fluctuations of 2.4 ± 0.4°C. These values were later used as reference to ensure natural light and temperature conditions during incubation experiments.

## Sample collection

Samples from four different species of benthic primary producers, each representing one of a major functional group in this backreef system, were collected using SCUBA. The investigated species comprised (1) a hermatypic coral, *Pocillopora damicornis*, (2) a crustaceous coralline red alga (CCA) *Hydrolithon reinboldii*, (3) a common fleshy macroalga in the backreef system, *Dictyota ceylanica*, and (4) a typical mixed consortium of turf algae. All specimens were collected from the backreef platform approximately 500–1000 m east of Paopao Bay (Cook's Bay) from water depths of 2.0–2.5 m and transferred in coolers to cultivation tanks within 1 h using watertight zippered polyethylene bags. Specimens were collected in replicates of at least 20. *Dictyota* specimens and CCA, growing as rhodoliths, were collected as whole individuals. For turfing algae pieces of reef structure at least 95% covered by the algae were collected. Fragments of *Pocillopora* colonies were collected using pliers, with each fragment from a different colony. *Pocillopora* fragments were then fixed onto ceramic tiles using small amounts of coral cement (Instant Ocean, Holdfast® Epoxy Stick) in such a way that only living coral tissue was exposed to the incubation waters. All samples were collected 5 days prior to the respective incubation experiments and incubated in common flow-through ambient water tanks to allow for healing of potential tissue lesions. Algal overgrowth on the ceramic tiles and glue junction was removed regularly and all specimens were carefully checked for potential infestation of epibionts or endolithic boring organisms to avoid potential confounding effects on experimental results. Samples of all primary producers were chosen in a way that they had comparable surface areas with an average of 87.3 ± 8.8 cm$^2$.

## Primary producer incubations
### DOC release

Each benthic specimen was incubated in an individual beaker following *Haas et al. (2011)* with minor modifications as follows. Primary producer incubations were conducted over a 24 h cycle to assess variations in daytime and night production rates. At dusk (1900 h) each specimen was placed into a randomized beaker containing 920 ml freshly collected filter sterilized backreef seawater (sterilized by passage through pre-flushed 142 mm polyethersulfone filters; 0.2 µm pore size). To measure initial DOC concentration, water was sampled from each beaker with an acid-washed HDPE syringe (60 ml) and filtered through a GF/F filter (Whatman; 0.7 µm nominal pore size) into precombusted glass storage vials with acid-washed Teflon septa. Parallel seawater controls were identical

with no organism added. Primary producers were incubated overnight 12 h and DOC samples were again collected at ∼0700 h of the following day. Primary producers were then incubated for another 12 h in the remaining 800 ml seawater. Final DOC samples were taken at around 1900 h. Specimens were then removed from the beakers using acid-washed forceps and the remaining incubation water was processed for the different exudate incubations (see below). Surface area and volume of all specimens (DOC and DO/pH incubations) were determined using the method described in detail by *Haas et al. (2011)*. Specimen volume was always <3% of incubation waters.

### Effects of primary producer physiology on DO

Parallel to the DOC release incubations, assessments of primary producer physiology on DO values of the surrounding water column were conducted. These incubations were set up simultaneously with and identical to the above described DOC incubation with the only difference being that beakers were sealed airtight with a low-density polyethylene film (Saran$^{TM}$) over the course of the experiment. The two parallel experiments allowed for reliable measurements of DOC release and algal physiology without the potential for contamination of DOC measurements across beakers (*Haas et al., 2011*). Initial DO readings were obtained from each beaker using a HACH LANGE HQ40 multiparameter instrument (DO: precision 0.01 mg l$^{-1}$, accuracy ±0.05%). DO readings were obtained in parallel to DOC sampling (1900 h, 0700 h, 1900 h) following the protocol described by *Haas et al. (2011)*.

## Exudate incubations

### Bacterial abundance and DOC fluxes

To resolve diel effects of different primary producer exudates on the ambient backreef microbial community and water chemistry, 48 h benthic and planktonic dilution culture incubations were conducted under natural light and in the dark. The remaining seawater from each replicate primary producer incubation beaker was filter-sterilized through a pre-flushed (1 L low-organic deionized water; Barnstead Nanopure) 0.2 μm polyethersulfone filter (Pall SUPOR-200) and then inoculated with (a) freshly collected unfiltered backreef seawater (100 mL incubation water : 40 mL inoculum) and (b) with freshly collected sand (15 mL sand : 125 mL incubation water) to add a compositionally-representative ambient planktonic or benthic microbial community to the sample exudate media, respectively. Of the four identical replicates generated in this way for each specimen one incubation vessel was then kept in the dark and a congruent sample under natural daylight conditions, both at *in situ* temperature over a time period of 48 h. Samples for DOC (DOC plus bacterial carbon) analysis and bacterial cell abundance were taken immediately after combining the exudates with the respective microbial inoculum and again at the end of the experiment. Bacterioplankton abundance was converted to carbon units assuming 20 fg C cell$^{-1}$ (*Lee & Fuhrman, 1987*) and total DOC concentrations in the exudate incubation cultures was calculated by subtracting bacterial carbon from the measured DOC. All organic carbon samples (30 mL) were collected in precombusted glass vials and immediately stored at −20°C for up to four months until analysis via high temperature catalytic oxidation

according to *Carlson et al. (2010)*. Additional samples for bacterial cell abundances (1–2 mL each) in the incubations were collected every 12 h over the 48 h incubation period. Bacterial abundance samples were immediately fixed in 0.5% paraformaldehyde and flash frozen at −80°C. Bacterial samples were stored for up to 2 months, and counted after 1X SYBR Green I (Invitrogen) staining via flow cytometry according to *Nelson et al. (2011)*. A parallel incubation with the remaining two congruent samples for each specimen was performed to determine DO fluxes. The remaining inoculated sub-samples were transferred to ground-glass stoppered bottles (Wheaton BOD) and initial oxygen concentration of each sub-sample was determined as described above. Samples were kept airtight under natural light conditions or in the dark at *in situ* temperature (26.6 ± 0.5°C) alongside DOC/microbial incubations. After 48 h DO values were measured again to assess microbially mediated oxygen fluxes.

### *In situ* benthic isolation tent deployment

Collapsible benthic isolation tents (cBITs) were used to assess effects of specific benthic communities *in situ*. The triangular pyramids, which were developed and built at the Smith and Rohwer laboratories, primarily consist of three transparent polycarbonate side panels joined by flexible polyvinyl chloride strips held erect by aluminum tubes which are connected by stainless steel cables. Tents were fixed to the ground with stainless steel pegs, with broad (25 cm) PVC flaps attached to the base of the tents to prevent water exchange, which were held flush to the sandy bottom with a 0.5-cm gauge stainless steel anchor chain. Tents enclosed a volume of approximately 0.12 m$^3$, and covered approximately 0.43 m$^2$ of the reef surface. All tents were equipped with autonomous recording data loggers (Manta 2, Eureka environmental engineering) that monitored temperature (precision 0.01°C), DO (precision 0.01 mg l$^{-1}$, accuracy ±1%, automatic temperature and pressure compensated and salinity corrected), and conductivity (accuracy ±1%, automatic temperature compensated) every 5 min over a minimum duration of 2 diurnal cycles (48 h). Further, tents were equipped with a circulation pump with tubing that recirculated water within the tents prior to sampling and, connected to a Niskin sampling bottle, served also as a sampling port (Fig. 1). Samples for DO, DOC, and bacterial abundance were collected every 24 h and processed as describe above. Five cBITs were set up simultaneously over 3 deployment periods ($n = 15$). Tents were mounted over selected benthic communities, comprising coral dominated, algae dominated, and sand dominated areas. Locations were chosen such that on each deployment period at least one representative of each of the respective benthic communities was included. After each deployment the benthic community enclosed by the cBITs was photographed for later determination benthic community composition. Photographs were processed using the digital image software Image J, allowing determination of the area occupied by the respective target organisms vs. the total projected area enclosed in each cBIT (Fig. 1).

### Data processing and derived variables

In all incubation experiments, rates of change in DOC and DO concentrations and, where applicable, bacterial abundances were calculated by dividing the difference between

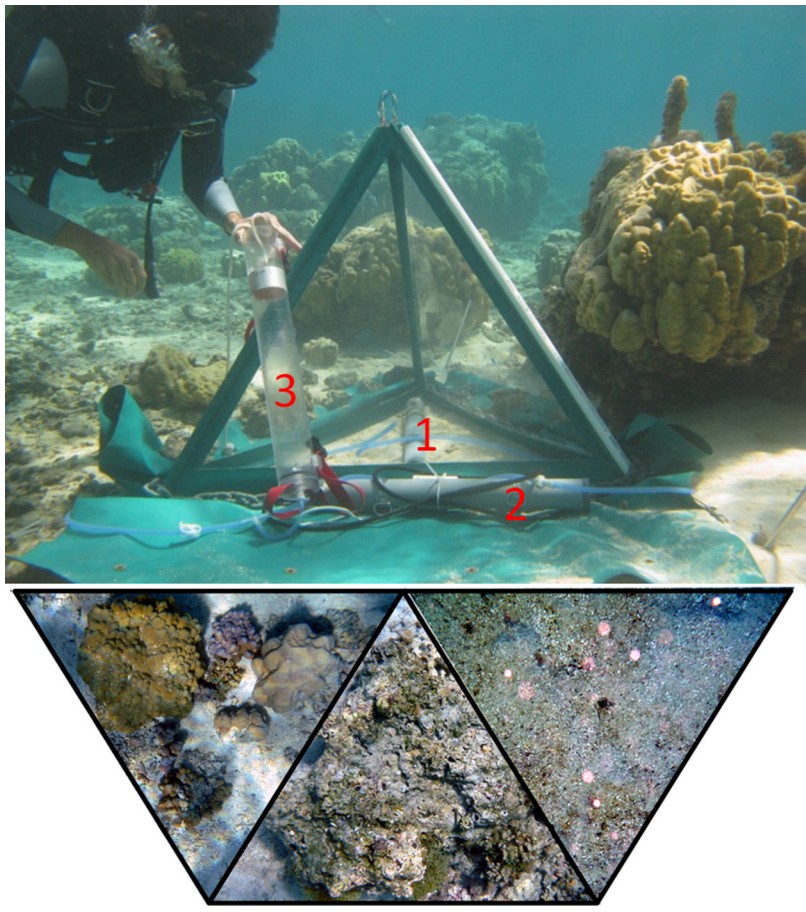

**Figure 1 Tent setup.** cBIT deployed *in situ* over different benthic communities (examples bottom left to right: coral dominated, algae dominated, sand) with data logger (1) and circulation pump (2) connected to a Niskin sampling bottle (3).

start and end concentrations by the incubation duration (12 h for primary producer incubations, 48 h for exudate incubations). The proportional release of photosynthates in primary producer incubations was calculated as the ratio of DOC:DO fluxes. Daylight rates of change in exudate incubations were calculated by subtracting the rate of change in analytes in the 48 h dark incubation experiments from the rate of change in analytes in 48 h incubations subjected to ambient day-night conditions (i.e., light = net − dark).

For incubations containing organisms or sand, rates of change in the assessed parameters were normalized per unit surface area of benthic material by dividing these rates by the surface area of the respective benthic organism or sand incubated in the starting replicate beaker (see *Haas et al., 2011* for surface area determination). In the case of the sand incubations, the rates of planktonic change were subtracted to account for water column effects. For all following microbial incubations, concentration independency was assumed, supported by the lack of correlation between rates of change in microbial oxygen consumption and DOC starting concentrations (least squares – Model I – regression

of oxygen fluxes on DOC starting concentrations: Benthic, $R^2 = 0.000152$; $p = 0.9451$; Planktonic, $R^2 = 0.018995$; $p = 0.3841$).

To estimate how different benthic assemblages influence DOC and DO availability and microbial activity, rates of change were extrapolated to the backreef community of our study site. To apply proportional impacts of the collective benthic and planktonic microbial community on the reef water column we assumed an average of 3 m water column depth containing the planktonic microbial community above a 7 mm thick layer of permeable calcareous reef sands containing the benthic microbial community. Sand permeability depth was calculated using a custom porewater sampler to (1) isolate a 45 cm diameter patch of sand, (2) withdraw sequential 2L volumes in which we (3) measure DO relative to overlying water; DO concentrations became hypoxic after 4L volume giving a permeability depth estimate of roughly 7 mm. This depth of exchangeable sand volume was replicated in the bottle experiments (15 mL sand across the 50 mm diameter glass bottle gives roughly 7 mm sand depth).

To assess rates of DOC concentration changes facilitated by the community enclosed in the respective cBITs, the difference between the concentrations in DOC samples collected at 0 and 24 h, were divided by the time between each the two sampling events (24 h). To calculate change in DO, sensor readings from deployment day 1 were subtracted from values measured at the exact same time point on deployment day 2. Hourly rates were then calculated by dividing the mean change from each measuring point from day one to day two by 24 (See Fig. S1).

Statistics were performed using SAS within the software package JMP (v10; SAS Institute 1989–2011). All statistical tests were conducted on log-transformed data to meet assumptions of normality. We tested whether derived rates differed from control treatments using analysis of variance (ANOVA) followed by Dunnett's post hoc test with $\alpha = 0.05$. To test whether rates differed among the treatments we used Tukey's post hoc tests with $\alpha = 0.05$. All rates are given in mean $\pm$ standard error (SE) where applicable.

## RESULTS

### Exudate release rates and photosynthesis of benthic producers

Mean rates of oxygen production during daylight by primary producers ranged from $24.5 \pm 0.9$ μmol L$^{-1}$ h$^{-1}$ (*Pocillopora*) to $35.1 \pm 2.5$ μmol L$^{-1}$ h$^{-1}$ (*Dictyota*). Nighttime oxygen consumption ranged from $5.7 \pm 0.6$ μmol L$^{-1}$ h$^{-1}$ (CCA) to $7.0 \pm 0.7$ μmol L$^{-1}$ h$^{-1}$ (turf). Rates of DOC release ranged from $0.9 \pm 0.3$ μmol L$^{-1}$ h$^{-1}$ (*Pocillopora*) to $5.5 \pm 1.8$ μmol L$^{-1}$ h$^{-1}$ (*Dictyota*) during daytime and $0.3 \pm 0.3$ μmol L$^{-1}$ h$^{-1}$ (CCA) to $1.2 \pm 0.3$ μmol L$^{-1}$ h$^{-1}$ (*Dictyota*) during night. Seawater control fluxes during both daylight and nighttime were significantly smaller for DO ($-1.08$ and $-0.57$ μmol L$^{-1}$ h$^{-1}$; Dunnett's $p < 0.001$) and were on average smaller for DOC ($0.29$ and $-0.28$ μmol L$^{-1}$ h$^{-1}$). Due to the small sample size, resulting from a loss of seawater control DOC samples, the latter was not statistically significant. All raw values of DO and DOC fluxes, facilitated by the primary producers are shown in Figs. S2 and S3. When using oxygen production as a

proxy for photosynthesis (*Haas et al., 2011*), the investigated primary producers released a mean of $12.6 \pm 2.5\%$ of their photosynthetically fixed carbon as DOC in the surrounding reef waters. On average, the fleshy macroalga *Dictyota* released the highest proportion ($21.6 \pm 3.4\%$) of the assimilated carbon followed by turf algae ($9.6 \pm 4.2\%$), CCA ($9.4 \pm 3.4\%$), and *Pocillopora* ($7.2 \pm 4.2\%$) as DOC.

In order to compare between effects facilitated by the different taxa, DO and DOC fluxes were corrected by normalizing the resulting values to the surface area of the respective organism. During daylight, oxygen production rates of primary producers significantly differed (ANOVA, $F_{4,20} = 27.9700$, $p < 0.0001$). Algae incubations showed significantly higher oxygen production rates than the seawater controls, whereby turf algae had the highest oxygen release rates per surface area of all incubated organisms (Tukey $p < 0.05$) (Fig. 2A). During dark incubations the primary producers exhibited significant differences in oxygen consumption rates (ANOVA, $F_{4,20} = 20.7842$, $p < 0.0001$). Congruent to daytime release rates, nighttime consumption was highest for turf, followed by *Dictyota*, CCA, and *Pocillopora* (Fig. 2A).

There were also significant differences in daytime DOC release rates between treatments (ANOVA, $F_{4,7} = 4.8069$, $p < 0.0350$). The non-calcifying alga *Dictyota* showed the highest release rates followed by turf, CCA, and *Pocillopora* (Fig. 2A). There was no statistically significant difference in DOC release rates detectable between the primary producers during dark incubations. However, similar to the daylight period, non-calcifying algae released, on average, higher amounts than the calcifying primary producers.

Over a whole diurnal cycle, net metabolic oxygen fluxes were significantly different between treatments (ANOVA, $F_{4,20} = 27.0087$, $p < 0.0001$). Turf algae had the highest rates of net oxygen production ($50.2 \pm 1.7$ mmol m$^{-2}$ d$^{-1}$), followed by *Dictyota* ($35.8 \pm 4.1$ mmol m$^{-2}$ d$^{-1}$) and the calcifying organisms CCA ($25.0 \pm 1.7$ mmol m$^{-2}$ d$^{-1}$) and *Pocillopora* ($15.6 \pm 5.3$ mmol m$^{-2}$ d$^{-1}$) (Fig. 3A). Similar differences were also detectable for net DOC release rates (ANOVA, $F_{4,6} = 16.6025$, $p = 0.0021$) whereby the fleshy alga *Dictyota* ($7.25 \pm 0.58$ mmol m$^{-2}$ d$^{-1}$) and turf ($4.30 \pm 0.72$ mmol m$^{-2}$ d$^{-1}$) had the highest diurnal DOC release rates followed by the calcifying organisms CCA ($2.28 \pm 0.60$ mmol m$^{-2}$ d$^{-1}$) and *Pocillopora* ($1.32 \pm 0.72$ mmol m$^{-2}$ d$^{-1}$).

## Microbial response

### Bacterial growth on organic matter released by benthic producers

There were significant differences in bacterial growth rates in both the benthic and pelagic communities depending on the source of the exudates (Fig. 4A) and the type of incubation (light vs. dark). Although the rate of bacterioplankton growth in the 48 h dark incubations was not significantly different between the exudate treatments (ANOVA $F_{4,19} = 1.2482$, $p > 0.05$), there were noticeable differences under daylight conditions (ANOVA $F_{4,18} = 5.9043$, $p = 0.0032$). Exudates from turf algae resulted in the largest increase in bacterioplankton growth followed by CCA and *Dictyota* (Fig. 4C). Coral exudates caused an average decrease in bacterial cells during daylight

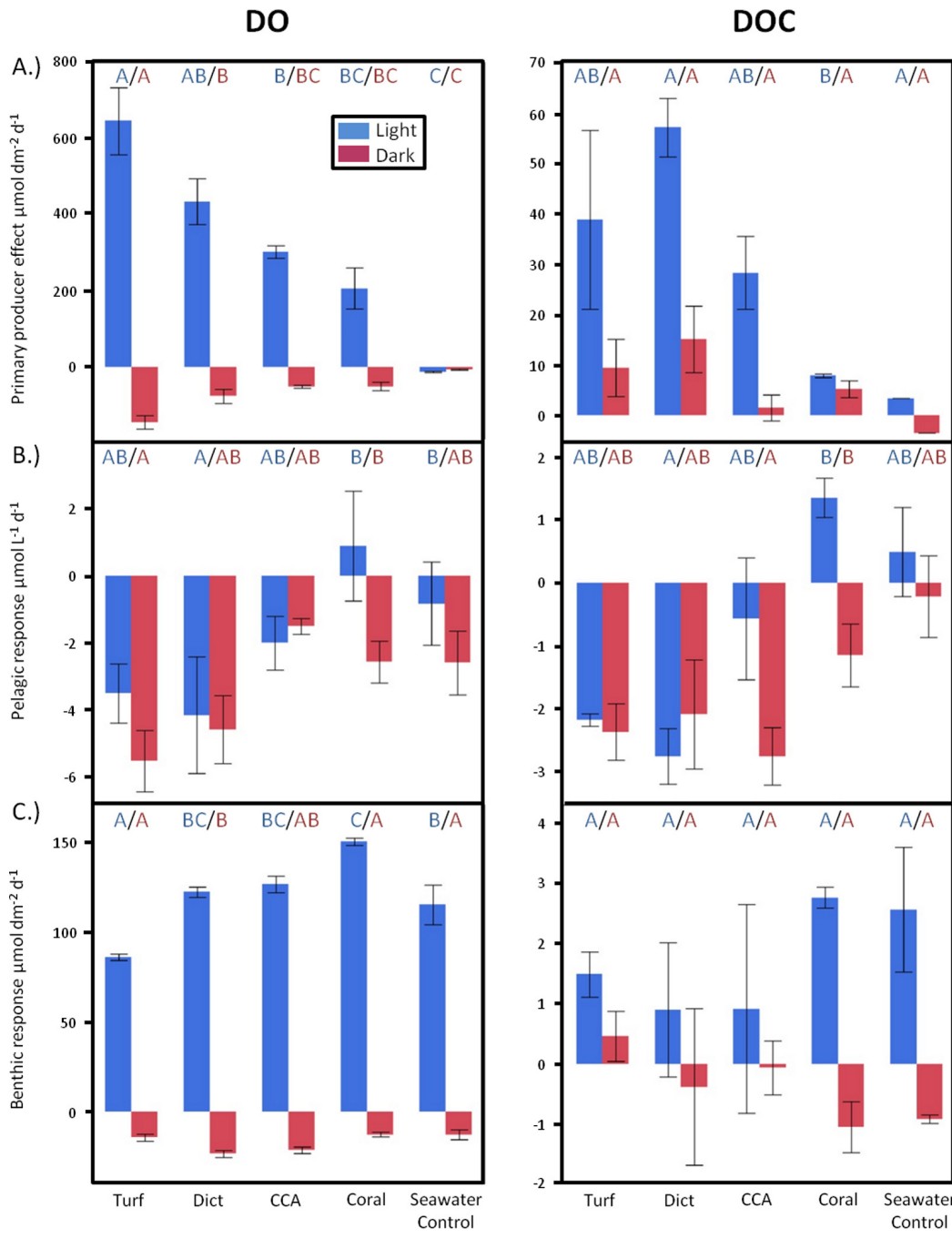

**Figure 2 Diurnally resolved responses.** Diurnally resolved direct effects of (A) different primary producer functional groups on surrounding DO and DOC concentrations (surface area corrected) and effects facilitated by (B) the pelagic and (C) benthic microbial communities as response to the respective exudates. Bars show mean values with standard error whiskers. Treatments with the same letter are not significantly different at $\alpha = 0.05$.

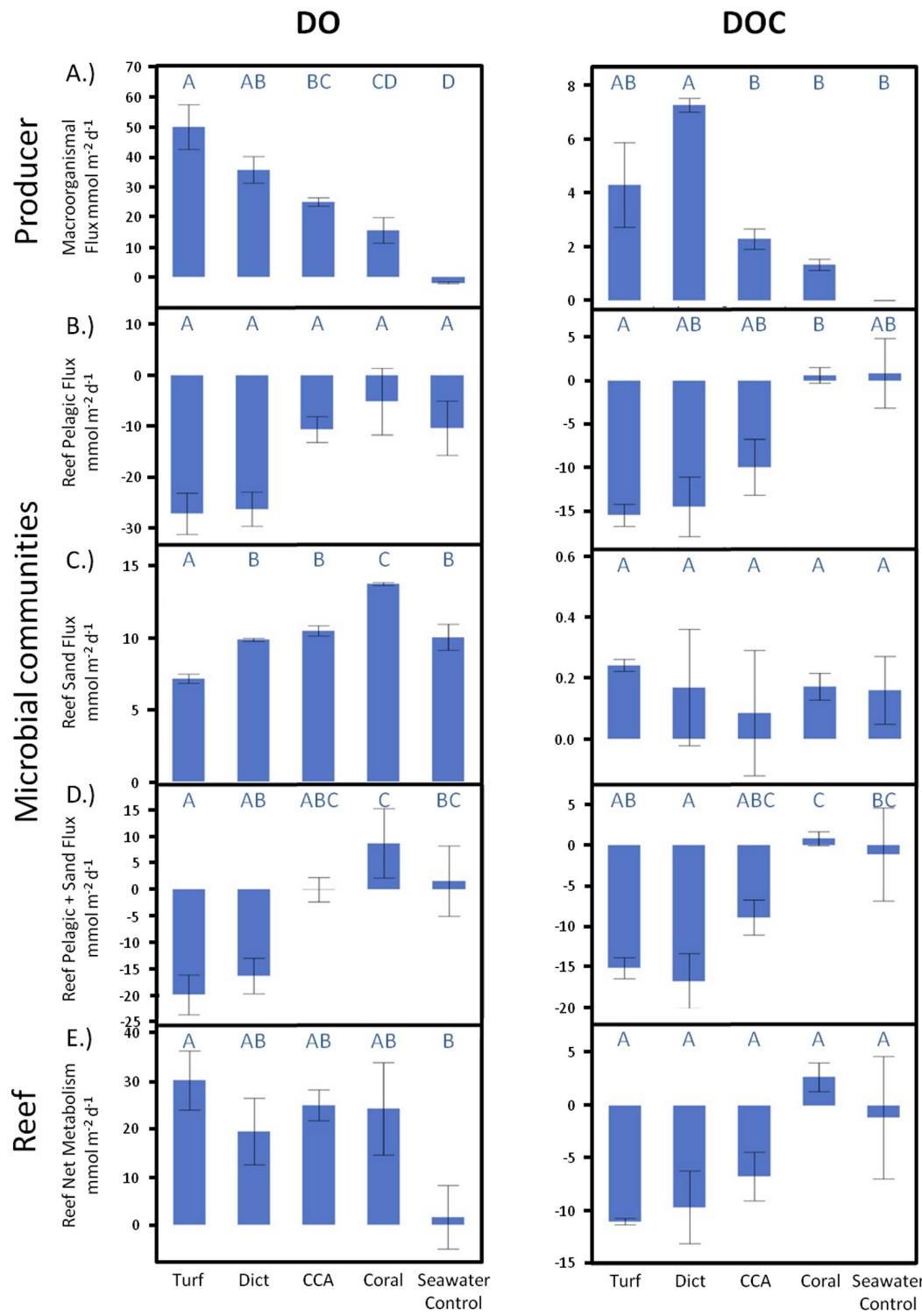

**Figure 3 Net responses.** Daily net fluxes of DO and DOC per m² reef area dominated by the respective primary producer functional group. (A) Changes facilitated directly by the 

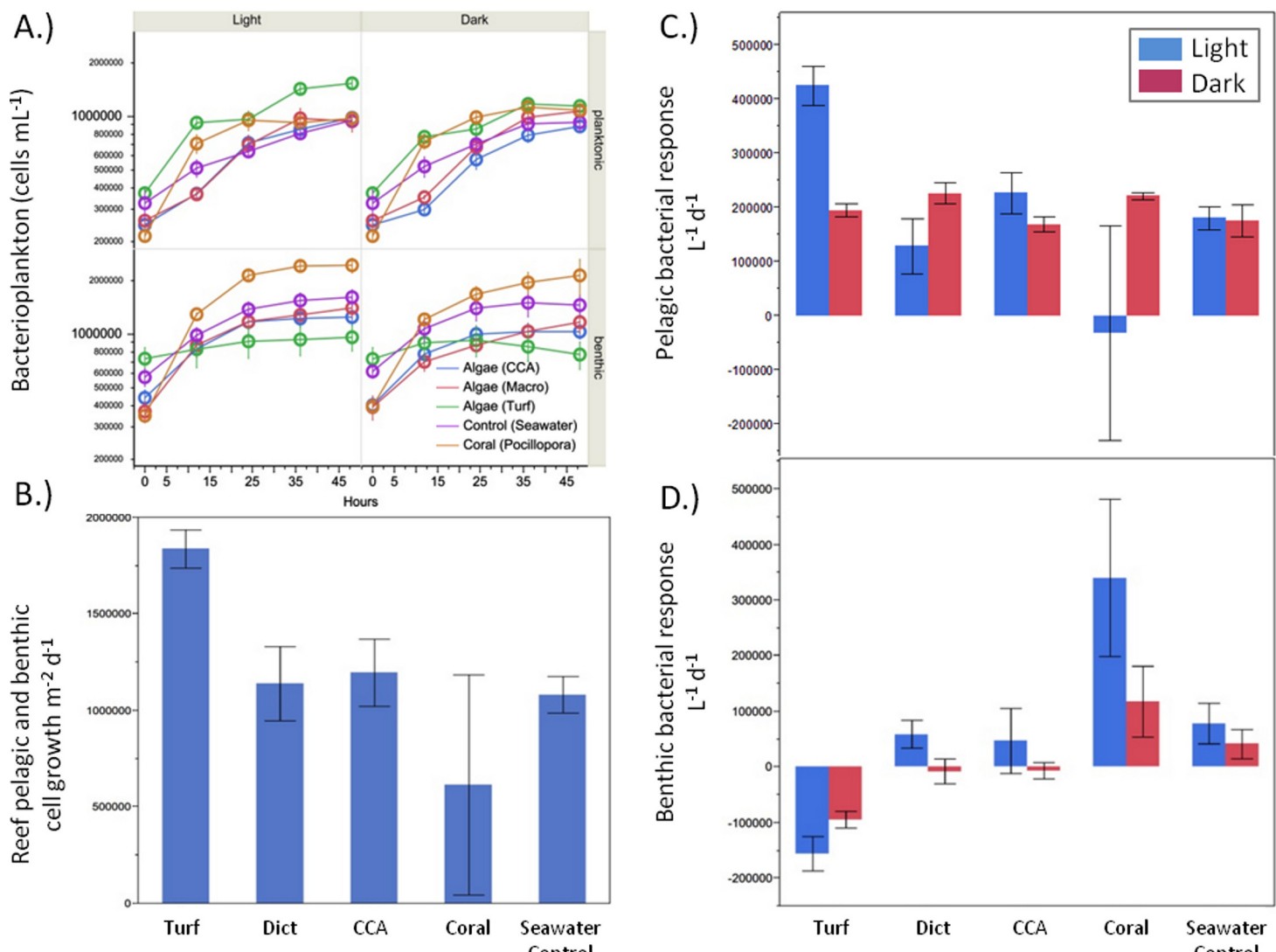

**Figure 3 (...continued)**
producers (B) Fluxes facilitated by the pelagic and (C) Benthic microbial community as a response to the respective exudates. (D) Combined pelagic and benthic microbial fluxes and (E) combined net producer and, responding to their respective exudates, microbially facilitated fluxes. Bars show mean values with standard error whiskers. Treatments with the same letter are not significantly different at $\alpha = 0.05$.

**Figure 4 Bacterial growth rates.** Bacterial growth rates in the respective exudate incubations. (A) raw values, (B) light phase resolved pelagic and (C) benthic response, and (D) combined pelagic and benthic growth over a whole 24 h time period.

hours. These patterns, where turf algal exudates increased and coral exudates decreased pelagic bacterial abundance, were also detectable over the whole diurnal cycle (ANOVA $F_{4,18} = 4.6153, p = 0.0090$), (Fig. 4B).

While the bacterial communities overlying the benthos responded differently to the exudate treatments in both the light (ANOVA $F_{4,17} = 8.7659, p = 0.0005$) and dark (ANOVA $F_{4,19} = 7.2631, p = 0.0010$), the response was opposite that of planktonic

bacterial community (Fig. 4D). Under both light and dark conditions, turf algal exudates decreased bacterial cell growth, while coral exudates caused the highest growth rate for the bacterial community associated with the reef sands (Fig. 4D). Accordingly, these patterns were also detectable when both rates of daylight and dark benthic bacterial growth were combined to calculate benthic bacterial growth rates over a full diurnal cycle (ANOVA $F_{4,18} = 16.7236, p < 0.0001$).

### Planktonic microbially mediated effects

Planktonic dilution cultures, in which different exudates were added to ambient bacterioplankton communities, showed significant differences in DO fluxes between the treatments both during daylight (ANOVA, $F_{4,19} = 4.2349, p = 0.0128$) and during dark (ANOVA, $F_{4,20} = 3.4412, p = 0.0270$) conditions. Exudates from the fleshy alga *Dictyota* and from turf algae resulted in the highest planktonic oxygen consumption rates both during dark and daylight conditions (Fig. 2B). While all treatments containing algal exudates and the seawater controls exhibited decreases in DO concentrations during the day, incubations containing DOM derived from the coral *Pocillopora* displayed an average increase in DO (Fig. 2B).

Significant differences were also detectable in daytime planktonic DOC consumption rates (ANOVA, $F_{4,13} = 4.6474, p = 0.0150$), where microbial communities growing on exudates derived from *Dictyota* and turf showed highest carbon draw down rates. As with DO fluxes, incubations with coral exudates were the only treatments that resulted in increases of DOC concentrations during daylight incubations (Fig. 2B). Even though there were no statistical significant differences in DOC changes of planktonic microbial dilution cultures between the treatments during dark incubations, in general, there was a trend whereby treatments containing algal exudates showed higher C drawdown rates than treatments containing coral exudates or seawater controls suggesting enhanced mineralization processes, data also supported by the oxygen concentration measurements (Fig. 2B). Over a whole diurnal cycle the planktonic microbial community had net consumption of DO and DOC (Fig. 3B) in all treatments where DOC algal exudates were amended. Coral exudate incubations, which yielded net increases in DOC indicates that photoautotrophy was stimulated via delivery of inorganic nutrients in coral exudate. Exudates of fleshy algae, and especially turf algae, resulted in significantly higher DO and DOC consumption rates than exudates derived from calcifying organisms (ANOVA, $F_{4,20} = 4.3867, p = 0.0104$ and ANOVA, $F_{4,13} = 4.9089, p = 0.0124$, respectively).

### Benthic microbially mediated effects

Benthic incubations, where primary producer exudates were amended to ambient microbenthos communities, showed significant differences in DO release rates among treatments during daylight incubations (ANOVA, $F_{4,19} = 9.5684, p = 0.0002$). Relative to control incubations, turf algae exudates resulted in significantly lower benthic microbial oxygen production, while the microbial community growing in coral exudate treatments showed a greater photosynthetic oxygen production rate (Fig. 2C).

Significant differences in DO consumption rates were seen between the treatments during dark incubations (ANOVA, $F_{4,19} = 5.2539$, $p = 0.0051$) (Fig. 2C), with the highest oxygen consumption rates for incubations containing exudates from the fleshy alga *Dictyota*. Algal exudate incubations showed generally lower DOC release rates during daylight hours than the benthic microbial communities growing on coral exudates or seawater controls. However, there was no significant difference in benthic microbe mediated DOC fluxes between the different treatments (Fig. 2C). However, algal exudate incubations showed generally lower DOC release rates during daylight hours than the benthic microbial communities growing on coral exudates or seawater controls.

Oxygen production rates of the microphytobenthos during the day living in the calcareous reef sands were always higher than the consumption rates during the night resulting in a net autotrophic metabolism over a whole diurnal cycle in all treatments (Fig. 3C). There were however significant differences between the treatments in net oxygen fluxes (ANOVA, $F_{4,19} = 13.6474$, $p < 0.0001$); turf algae exudates significantly decreased the productivity of the benthic microbial community, while coral exudates significantly increased microbial productivity. Exudates derived from macroalgae and calcifying algae had no significant effect on oxygen production rates of the microbenthic communities growing on sand (Tukey $p > 0.05$). All treatments showed DOC release of the benthic microbial community over a full diurnal cycle, though there were no statistically significant differences detectable between among the treatments (Fig. 3C).

### *Combined microbial effects*

When extrapolated to a landscape scale within the lagoon on Mo'orea's backreef, the combined benthic and planktonic microbial metabolism in the seawater controls was net autotrophic during the daylight hours and heterotrophic during the night. These rates were roughly equal, thus over a diurnal cycle the combined microbial community metabolism (planktonic net heterotrophic and benthic net autotrophic) was balanced with neither DO nor DOC concentrations changing noticeably over 24 h (Fig. 3D). The addition of exudates had significant effects on the net microbial community metabolism for DO (ANOVA, $F_{4,19} = 5.7686$, $p < 0.0033$) and DOC (ANOVA, $F_{4,8} = 7.7374$, $p < 0.0074$) fluxes. Exudates derived from non-calcifying algae, especially turf, significantly increased oxygen (Dunnett's $p < 0.0299$) and DOC (Dunnett's $p < 0.0467$) consumption rates as a whole with an increase in planktonic microbial consumption and a decrease in benthic microbial production (Figs. 3D and 5). In contrast, the combined microbial community metabolism resulted in increased net oxygen production in the presence of coral exudates, facilitating increases in DO and DOC concentrations of the surrounding water column. The shift towards an increasingly autotrophic microbial community in coral exudate incubations was due to reduced consumption rates by the planktonic microbial community and increases in the productivity of the benthic microbial community (Figs. 3B–3D).

### Community wide (microbial + macrobial) effects

Comparison of the extent of DOC and DO change facilitated by primary producer exudates and subsequent microbial metabolism showed that the high oxygen production rates

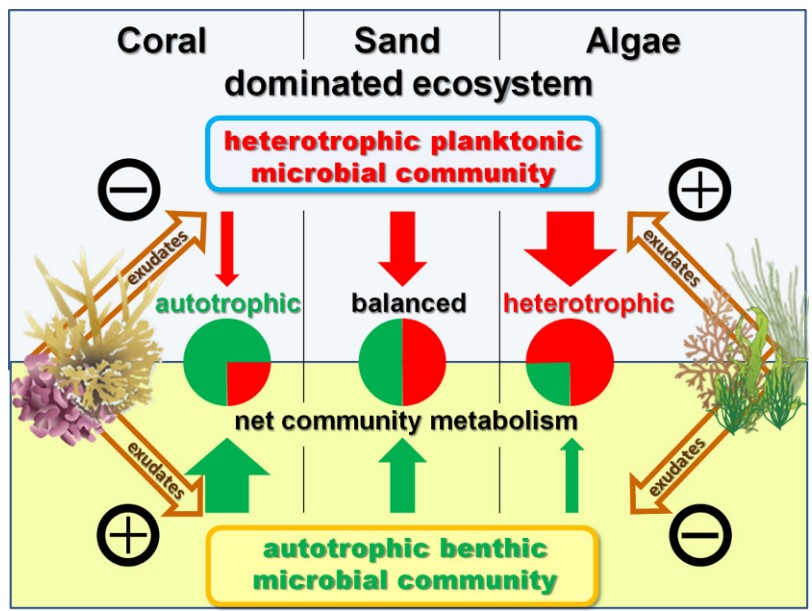

**Figure 5 Exudate influences.** Schematic model of exudate influences on benthic and pelagic microbial community metabolism. Without exudates present the net heterotrophic pelagic and net autotrophic benthic microbial community exhibit comparable magnitudes in their opposing influences on DO availability. Coral exudates facilitate a shift towards a net autotrophic system by increasing the net production of the benthic and decreasing the net consumption of the pelagic community. In contrast, algal exudates facilitate a shift towards a net heterotrophic system by decreasing the net production of the benthic and increasing the net consumption of the pelagic microbial community.

of primary producers exceeded consumption by the combined planktonic and benthic microbial communities (Fig. 3E), maintaining overall net autotrophy in the system. Despite the elevated $O_2$ production for the non-calcifying algae (Fig. 3A) there was sufficient stimulation of heterotrophic bacterial respiration (Fig. 3B) such that no resolvable difference could be resolved in the net metabolism between the amended treatments (Fig. 3E).

In contrast to dissolved oxygen dynamics, exudates from non-calcifying algae strongly increased planktonic microbial DOC consumption rates to values that exceed the release rates of the exudates, leading to a net removal of DOC in turf and macroalgae (at least of the species measured here) dominated reef environments (Figs. 3E and 6). Estimating rates of DO and DOC change in the investigated lagoons of Mo'orea based on reported benthic cover data (Turf, 35%; Macroalgae, 35%; Coral, 22%; Sand, 8%, see Methods above) yielded a net excess of oxygen production ($+7.6 \ \mu mol \ L^{-1} \ d^{-1}$) and net consumption of DOC ($-2.2 \ \mu mol \ L^{-1} \ d^{-1}$). However, these broad extrapolations should be treated with caution as there may be important species specific rates of production and DOC release within each of the benthic functional groups that we have yet not accounted for.

## In-situ mesocosm fluxes

To evaluate the calculations obtained from the beaker incubations, we compared our estimates of reef-scale ecosystem fluxes with results from *in situ* mesocosm incubations using cBITs. DO and temperature loggers deployed within each of these semi-closed

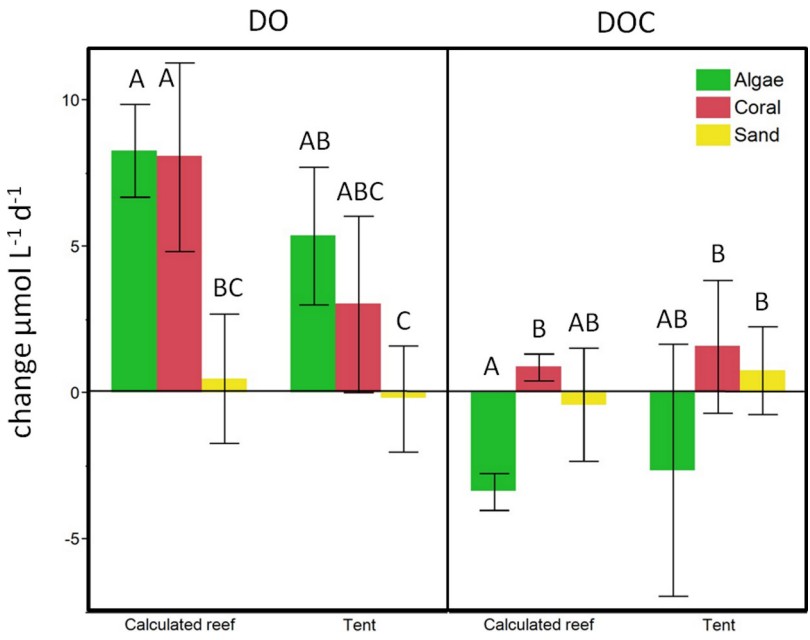

**Figure 6 Beaker incubation to tent comparison.** Calculated change for the respective benthic cover on a lagoonal scale compared to *in situ* cBIT community metabolism measurements. Bars show mean values with standard error whiskers. Treatments with the same letter are not significantly different at $\alpha = 0.05$.

mesocosms showed that DO values significantly increased in all the tents containing coral (pairwise two-tailed t-test $p < 0.0033$) or turf algae (pairwise two-tailed t-test $p < 0.0001$) over a 24 h period. Tents deployed on sand; however, showed no statistically significant changes in oxygen concentrations over a 24 h interval (Fig. 6). DOC concentrations increased on average in tents deployed above corals and sand flats and showed an average decrease in tents enclosing algae-dominated communities. The magnitude and direction of DO and DOC fluxes matched those derived from extrapolations based on the bottle incubation experiments described above (see Fig. 6).

## DISCUSSION

Our study reports for the first time how the responses of different microbial compartments (benthic and planktonic) to benthic primary producer exudates contribute to the reef-scale community metabolism of oxygen and DOC. By coupling these responses with measurements of photosynthesis and exudate release by benthic producers, we are able to model the influence of benthic community structure on reef community metabolism. Previous work has described how antagonistic coral reef associated ecosystem engineers (*Berkenbusch & Rowden, 2003*) like coral and non-calcifying algae have the potential to alter biogeochemical cycling and microbial processes (*Wild et al., 2011*). While previous work has measured rates of benthic primary production (e.g., *Wanders, 1976*; *Hatcher, 1990*), organic matter release by benthic producers (e.g., *Brylinsky, 1977*; *Tanaka et al., 2008*; *Muscatine et al., 1984*) and subsequent exudate remineralization by the planktonic microbial community (e.g., *Ferrier-Pagès et al., 2000*; *Haas et al., 2011*), we believe our

study is the first to address the cumulative effects of these processes over diurnal cycles and at a landscape scale to holistically estimate the effects of benthic primary producers on reef metabolism.

Similar to previous studies (*Hatcher, 1990*; *Wild et al., 2010*; *Haas et al., 2011*), the algae investigated here (particularly the non-calcifying taxa) showed significantly higher rates of net primary production than the hermatypic coral. Concomitantly, the fleshy algae fixed more inorganic carbon during photosynthetic processes (estimated as higher DO production) and also exuded significantly higher amounts of DOC into their surrounding environment. DOC release rates of the different primary producers (8–57 $\mu$mol dm$^{-2}$ d$^{-1}$) were comparable to release rates previously published from reef locations around the world (*Naumann et al., 2012*, Red Sea corals: 6–56 $\mu$mol dm$^{-2}$ d$^{-1}$; *Haas et al., 2010b*, Mexican Caribbean algae: 28–208 $\mu$mol dm$^{-2}$ d$^{-1}$, *Haas et al., 2011*, Central Pacific coral and macroalgae: 48–336 $\mu$mol dm$^{-2}$ d$^{-1}$). In the present study, over a full diurnal cycle, benthic primary producers released about 10% of their daily fixed carbon as DOC in the surrounding waters.

Responses of the associated microbial communities to these exudates varied widely and were dependent on the source of the exudates as well as the habitat that the microbes originated from. Contrary to previous studies which generally linked the abundance of highly productive macroalgae to an overall increase in net metabolic balance (*Wanders, 1976*; *Hatcher, 1990*), our study indicates that, as a result of subsequent increased remineralization of exudates by planktonic and benthic microbial communities, the estimated net oxygen production does not vary significantly between coral and algae dominated reef systems on a community scale. Further, our results suggest that, with shifts from coral to algae dominated systems, dissolved organic carbon concentrations in the water column will decrease as a result of an elevated heterotrophic microbial community metabolism, congruent with demonstrated DOC depletion in shallow reefs (*Nelson et al., 2011*).

Results from the beaker incubations containing either benthic or planktonic microbes and seawater only showed that while the planktonic microbial community was consistently net heterotrophic the benthic microbial community metabolism was net autotrophic due to daytime photosynthesis, producing significantly higher amounts of oxygen during the daylight hours than it consumed over a 24 h period. Scaled volumetrically to the scale of a 3 m deep reef ecosystem, the effects of the respective net autotrophic benthic and net heterotrophic planktonic microbial communities had comparable magnitudes, resulting in a combined neutral net microbial community metabolism with no significant change of DOC and DO values over a whole diurnal cycle.

The introduction of exudates, however, had noticeable and significantly diverging influences on this balanced community metabolism. Coral exudates increased the net planktonic microbial community production, changing the net oxygen production towards an average positive balance during daylight hours. Coral exudates also enhanced the inherently autotrophic character of the microphytobenthos, such that at the reef scale coral exudates overall stimulated net ecosystem productivity (Fig. 5). Significant increases in the abundance of autotrophic microbes as a result of available coral exudates have been

previously reported (*Ferrier-Pagès et al., 2000*). Although the autotrophic cells do not necessarily rely on organic nutrients, it has been demonstrated that they can also take up DOM, such as amino acids, as an inorganic nutrient source (*Flynn & Butler, 1986*; *Palenik & Morel, 1990*). Autotrophic microbial biomass enhancement may thus be mediated by an increase in bioavailable inorganic nutrients, supplied by heterotrophic remineralization of coral exudates in the biocatalytic reef sands (*Szmant, Ferrer & FitzGerald, 1990*; *Schlichter & Liebezeit, 1991*; *Ferrier-Pagès et al., 2000*; *Wild et al., 2004a*; *Wild, Woyt & Huettel, 2005*). In contrast, addition of algal exudates, most noticeably exudates derived from turf algae, stimulated heterotrophic oxygen and organic carbon consumption rates by the planktonic and benthic microbial community, mediating an overall shift toward a significantly more heterotrophic microbial community metabolism. Although the changes in the combined microbial community metabolism did not outweigh the high oxygen production rates of benthic algae, the exudates did result in increased microbial DOC consumption which exceeded measured DOC exudation rates (Figs. 3A–3E). When scaled to the 3 m deep reef ecosystem this imbalance resulted in a net bacterial carbon demand which exceeded the rate of carbon exudation by algae, suggesting a possible mechanism for observed depletion of DOC in waters overlying the reef relative to oceanic inputs (*Nelson et al., 2011*). Our previous study conducted in this reef system demonstrated that exudates from fleshy macroalgae were enriched in specific carbohydrate components and were more labile than exudates derived from corals, fostering rapid but inefficient growth of primarily copiotrophic bacterioplankton in the surrounding water column. By facilitating the remineralization of semi-labile DOC inputs from the open ocean (sensu *Carlson et al., 2002*) the high carbon demand of inefficient copiotrophic "super-heterotrophs" (*Dinsdale & Rohwer, 2011*) may be a mechanism fueling the excessive carbon consumption rates estimated here and the subsequent depletion of DOC (*Nelson et al., 2011*) on reefs dominated by fleshy algae such as the backreef of Moʻorea.

In contrast, the shift towards a net autotrophic metabolism of the collective microbial community stimulated by coral exudates likely compensates for the initially lower photosynthetic oxygen production rates of corals compared to algae (*Wanders, 1976*; *Hatcher, 1990*). In our estimates this resulted in comparable net oxygen fluxes of the combined community metabolism in coral compared to algae dominated locations. Coral exudates facilitated changes in the microbial community metabolism towards higher primary production rates and led to an overall increase in DOC concentrations (resulting from net coral and microbial DOC release). Together these results suggest that reefs dominated by corals, by stimulating microbial primary production, may maintain comparable net ecosystem productivity to those dominated by fleshy algae, but additionally may maintain elevated levels of potentially labile DOC available for remineralization and recycling by microbial communities.

Our reef-scale estimates of benthic productivity and benthic and pelagic microbial metabolism from habitat-specific bottle incubations were validated by *in situ* measurements using contained benthic incubation tents (cBITs; Fig. 1); net metabolic balances of both oxygen and DOC assessed in benthic tents over a whole diurnal cycle showed

the same trends as suggested by the reef wide calculations derived from our incubation experiments (Fig. 6). This study may thus suggest potential explanations for previously observed discrepancies of primary producer effects on *in situ* values of DO and DOC concentrations. For example, while some studies identified higher net primary production, and concomitant $O_2$ production, of fleshy algae compared to corals (*Wanders, 1976*; *Hatcher, 1990*; *Done, 1992*; *Haas et al., 2011*), *in situ* measurements revealed lower oxygen concentrations in algae dominated areas compared to reef locations with high cover of hermatypic corals or other calcifying organisms (*Haas et al., 2010b*; *Niggl, Haas & Wild, 2010*). This is also the case for the proposed effects of the primary producer communities on surrounding DOC concentrations. While multiple studies in various coral reef systems have shown that coral reef associated primary producers, and particularly fleshy macro- and turf algae, release a noticeable portion of their photosynthetically fixed carbon as dissolved material into their surroundings (*Wild et al., 2010*; *Haas et al., 2011*), recent *in situ* assessments of tropical reef environments identified significantly lower DOC concentrations associated with higher algal abundance (*Dinsdale et al., 2008*; *Nelson et al., 2011*) throughout the tropical Pacific. This apparent discrepancy has been suggested to be caused by a co-metabolism of refractory carbon that occurs when microbes are given an excess labile carbon (*Carlson et al., 2002*; *Dinsdale et al., 2008*; *Nelson et al., 2011*). The differences between the direct influences of the different primary producers on oxygen and DOC (i.e., high production rates of non-calcifying algae) and the resulting values measured *in situ* values (low DO/DOC concentrations associated with high algal cover) may be explained by the unaccounted-for influences of algal and coral exudates on collective microbial metabolism demonstrated here.

Beyond the direct effects of primary producers and the indirect effects of microbes on key environmental parameters (DO, DOC), this study also shows noticeable influences of specific primary producer exudates on the community metabolism and the abundance of microbes in the different reef habitats (benthic vs. planktonic). While turf algal exudates led to significant increases in the abundance of the collectively net heterotrophic microbial community, they simultaneously mediated a significant decrease of bacterial growth rates in the generally net autotrophic benthic environment. Coral exudates in contrast showed no considerable effect on cell abundance in the planktonic environment, but fostered significantly higher growth rates than all other treatments in the predominately autotrophic benthic associated microbial community.

A possible caveat of our study is that we assumed concentration independency on influences of the different exudates on the microbial communities. Although the data supports this assumption partly as there is no direct relation between the starting DOC concentrations and the microbial oxygen draw down rates, and a related study (*Nelson et al., 2013*) showed that the composition of the respective exudates has a significant influence on the microbial community, there may be concentration thresholds at which the overall amount of the organic carbon supplied to the system will play a role in microbial metabolism. Nevertheless, the fact that the contrasting effects of the exudates derived from coral and algae change the metabolism in the planktonic and benthic environment relative

to the controls regardless of the amount of exudate used suggests that the mechanisms described above are of ecological significance. Additional validation of these mechanisms is given with the data collected from the *in situ* cBITs.

## CONCLUSION

This study primarily highlights the variability of benthic primary producer influences in different ecosystem compartments. It indicates their diverging effects on planktonic and benthic microbial ecology and subsequently on biogeochemical resources. The shift from net autotrophic towards net heterotrophic microbial community metabolism, accompanying changes from coral to algal dominance, may thereby have potential negative implications on energy transfer to higher trophic levels (*McDole et al., 2012*). Our results suggest that the bioavailable energy (DOC) provided by the macrobial photosynthetic organisms will not be able to support the multitude of trophic levels found in this otherwise oligotrophic coral reef environment, but rather fuels a short linked and inefficient (*Haas et al., 2011*; *Nelson et al., 2013*) microbial metabolism. Finally, our use of controlled incubations, coupled with *in situ* mesocosm experiments, provides the first comprehensive view of benthic primary producer-induced, and microbially-mediated alterations of biochemical cycling over diurnal cycles in both benthic and pelagic shallow reef environments. While it is uncommon for researchers to simultaneously assess the independent and combined contribution of macro- and micro-organisms to reef-scale metabolism, our results suggest that this approach will be necessary if we are to accurately predict how reef communities will change in response to the multitude of global and local stressors currently impacting them.

## SUMMARY

The following collection of studies supports the idea that the microbial and macrobial community are strongly interrelated and subjected to positive feedback loops in which contribute to phase shifts from coral to algal dominance.

Microbes inhabiting coral surfaces are subjected to shifts in community composition and elevated activity in response to increased availability of algae derived DOM. This increased microbial activity, facilitated by bioavailable algae derived OM, has been identified as a key mechanism leading to coral mortality. First, *Kuntz et al. (2005)* and *Kline et al. (2006)* showed that elevated concentrations of organic compounds were more detrimental to coral health than increased availability of inorganic nutrients. Concurrently, *Smith et al. (2006)* conducted an empirical study which demonstrated that coral mortality was mediated by algal released dissolved compounds which induced microbe facilitated hypoxia. Supporting results have been provided by *Barott et al. (2009)* and a recent study by *Morrow et al. (2012)* who identified consistent patterns in physiology and microbial community differentiation across different types of coral-algal competitive interactions. Here turf- or macroalgae interactions with corals created a zone of hypoxia and altered pigmentation in the coral tissue. In the companion articles (*Haas et al., 2010a*; *Haas et al., 2010b*) we can provide the first direct visualization of oxygen gradients originating from corals and algae, as well as at the interfaces. The study shows 2 dimensional images of

oxygen gradients over time in varying flow conditions and can thereby provide compelling evidence for the existence of hypoxic zones in coral-algae interaction processes. By using oxygen optodes as biological sensors, *Gregg et al., 2013*, revealed that in these processes the source of DOC, rather than the microbial community is the driving factor for microbial oxygen drawdown.

On a larger scale *Dinsdale et al. (2008)* described an increase of the microbial density by an order of magnitude from islands dominated by hermatypic corals and coralline algae towards islands dominated by fleshy macro- and turf algae. This study also demonstrated that on islands with high cover of fleshy macro- and turf algae the microbial community was dominated by heterotrophs, including a large percentage of potential pathogens. Adding to this, *Nelson et al. (2013)* showed that, in contrast to coral exudates, which facilitated high microbial diversity with few virulence factors, macroalgal exudates selected for less diverse communities heavily enriched in copiotrophic lineages, containing pathogens with increased virulence factors. These copiotrophic lineages, which are selected for in energy-rich surroundings resulting from labile, algae derived organic material have been previously described as "super-heterotrophs" (*Dinsdale & Rohwer, 2011*).

This shift towards higher abundance of planktonic microbes has been described in a recent study by *McDole et al. (2012)* as "Microbialization". Here, a microbialization score was applied to each site representing the percentage of the microbial metabolic energy consumption opposed to macro-organism facilitated energy fluxes in an ecosystem. A survey of 99 locations across the tropical Pacific demonstrated a strong correlation between these reef microbialization scores and human impact. In impacted systems *McDole et al. (2012)* identified a reallocation of bioavailable energy, provided by the primary producers from more complex organisms (e.g., fish biomass) to microbes. While T McDole, unpublished data (companion manuscript) suggest that in stages of intermediate degradation (noticeable, but limited human impact, e.g., Molokai, Hawaii/Saipan, Northern Mariana Islands) a disproportionately high amount of energy is dissipated by autotrophic microbes, they further suggest that in severely degraded reef systems (high human impact, e.g., Oahu & Niihau, Hawaii), pathways of energy flow are reestablished through heterotrophic microbes with more pathogen-like or copiotrophic growth strategies.

Collectively, these studies show that in coral reef systems, organic matter dynamics are tightly coupled with the benthic primary producers and the associated microbial community. Further it becomes evident that these factors strongly influence each other and may create positive feedback loops by (a) fostering ineffective and more pathogenic microbial communities, which (b) facilitate regions of decreased oxygen availability through metabolic activities, and (c) alter the transfer of energy to higher trophic levels, during shifts from coral to algal dominance on tropical reefs.

## ACKNOWLEDGEMENTS

We would like to give special thanks to Dr. Nichole Price for her help with the design and construction of the collapsible benthic isolation tents (cBIT).

### Funding

This research was supported by the National Science Foundation (NSF; http://www.nsf.gov/) award OCE-0927415 to JES and FLR and OCE-0927411 to CAC. Funds were provided to investigate the coupling between DOM, algae, and microbes on coral reef platforms. The funders had no role in study design, data collection and analysis, decision to publish, or preparation of the manuscript.

### Grant Disclosures

The following grant information was disclosed by the authors:
National Science Foundation (NSF; http://www.nsf.gov/): OCE-0927415, OCE-0927411.

### Competing Interests

There are no competing interests.

### Author Contributions

- Andreas F. Haas, Craig E. Nelson, Linda Wegley-Kelly and Steven D. Quistad conceived and designed the experiments, performed the experiments, analyzed the data, wrote the paper.
- Forest Rohwer, Craig A. Carlson and Jennifer E. Smith conceived and designed the experiments, performed the experiments, analyzed the data, contributed reagents/materials/analysis tools, wrote the paper.
- James J. Leichter conceived and designed the experiments, performed the experiments, contributed reagents/materials/analysis tools, wrote the paper.
- Mark Hatay conceived and designed the experiments, analyzed the data, contributed reagents/materials/analysis tools, wrote the paper.

### Supplemental Information

Supplemental information for this article can be found online at http://dx.doi.org/10.7717/peerj.108.

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
