# Peer review of "Influence of coral and algal exudates on microbially mediated reef metabolism"

_PeerJ, doi:10.7717/peerj.108_

## Round 0.1 · original submission · Minor Revisions

Please follow the comments by both reviewers which I believe will help improve your manuscript.

·

Basic reporting

Very nicely written introduction; incorporating both seminal and more recent literature on the topic.

Just a few minor comments on Figures:
Figure2 – Please state in legend how error bars were calculated, and remind readers that A/A, A/B etc. are representative of Tukey post-hoc statistical significance results. Also, please state in your results section whether your +/- error are SD or SE (see pg 11)

Figure5 – Can the positive and negative signs be moved above and below the exudate arrows? This is your call in the end, it just looks a little bit busy and hard to interpret as it is. I like the concept though.

Figure6 – What level of water flow is calculated into this model? If any, can you mention in the figure legend?

Minor note: Rates is listed twice in the last sentence of the first ending paragraph on pg. 18

Experimental design

The experimental design was clearly conceived and implemented. The authors made every effort to control for confounding influences on complex microbial processes related to the different benthic and pelagic components they examined. I have no additional comments.

Validity of the findings

The discussion does a nice job of summarizing a complex story, however it is a bit long and overly detailed in spots. Go over the discussion once more and try to tighten up the wording and reduce any redundancy in the discussion of your results. Otherwise, this was a very enjoyable paper to review and I appreciate the time and effort you clearly put into the design, implementation, and writing of the study.

Additional comments

No additional comments

Reviewer 2 ·

Basic reporting

General edits:
Throughout the document the formatting shifts from double spaces between paragraphs to indented paragraphs – this needs inspection and revision for consistency
Pg 9 ln 32 (last line) remove one of the double used ‘the’
Pg 10 ln 7 correct spelling ‘subracted’ to subtracted
Pg 16 Para 2 ln 8 – ‘further’ used twice – reword
Pg 16 Para 2 ln 9 – ‘watercolumn’ should be two words
Pg 18 Para 1 ln 7 – remove the first ‘rates’
Pg 19 Para 1 ln 5 – surrounding should be surroundings
Pg 19 Para 1 ln 8 – pacific should be Pacific
Pg 19 Para 2 ln 2 – (DO, OC), I think should be (DO, DOC)?
Pg 19 Para 3 ln 6 – ‘thee’ should be ‘their’
Pg 20 Para 3 ln 5 - pacific should be Pacific

Introduction
Acceptable

Methods
Statistical methods used in the data analyses should be included.

Results
Acceptable

Discussion
The authors have provided very interesting results; however, for the first three pages of the discussion the authors just seem to reiterate their results and overall the discussion does not flow smoothly. For example, helpful interpretations or explanations of the significance of a particular point would help carry the reader from one point to the next and would improve the flow.
Reviewer suggests that the discussion could be improved by developing the significance or ‘so what’ aspect more thoroughly and explaining the ecological consequences of their statements, particularly since Peer J has a broad audience.

Perhaps the authors are relying on previous publications on related experiments (e.g., Haas et al. 2011) for details? That publication read well and was easy to follow the logic and story-line. It may be helpful for the authors to compare this manuscript to the previous.

For example:
Pg 17 Para 4 – ‘On the contrary....’ this is a single sentence, not a paragraph, this needs further development such as explaining the consequence of shifting to a significantly heterotrophic microbial community metabolism.

Pg 18 Para 1 – this paragraph could be improved by explaining further the possible consequences

Pg 18 Para 2 – This final statement, seems to beg for further explanation as to ecological value of this situation renders.

Pg 20 Para 1- ‘above described mechanisms are of ecological significance....’ This is a good example of providing further explanation or examples defining the significance.


Figures
Fig. 2 – legend, correct spelling ‘functional’
Fig 2 – legend, reword #3 ‘communities as response’ suggestion: ‘communities as a response’

Fig 4 legend and graphs – Please review this Figure the legend has letter designations and the graphs have number designations. Also the graph’s order does not seem to match. Were statistical differences intended to be shown on the bar graphs?

Fig 6 – The text indicates statistics were done, but the graphs do not reflect any statistical differences as in the other graphs, was this intended?

Experimental design

no comments

Validity of the findings

No Comments

Additional comments

Just to summarize, I think the authors provide interesting and valuable information, but the message does not seem as well developed and as clear as in several of the previous articles by this group. I think the readability of the article could be improved if the authors considered they were writing to a broad audience.

---

## Round 0.2 · accepted · Accept

Thank you for carefully addressing the reviewers' feedback. I believe the manuscript has been strengthened by their input.